

# Antibacterial potential and chromatographic profiling of bioactive compounds from endophytic *Streptomyces* sp. strain MIRK71 isolated from *Mirabilis jalapa* (L.)

Lalrokimi[1], Purbajyoti Deka[1], William Carrie[1], Lallawmsangi[1], Christine Vanlalbiakdiki Sailo[2], Lalrosangpuii[3], Felicia Lalremruati[1], Awmpuizeli Fanai[1], Yasangam Umbon[4], Esther Lalnunmawii[1] and Zothanpuia[5]

[1] Department of Biotechnology, Mizoram University, Aizawl, Mizoram, India
[2] Microbiology Department, Synod Hospital, Durtlang, Aizawl, Mizoram, India
[3] Department of Life sciences, Pachhunga University College, Aizawl, Mizoram, India
[4] Department of Pharmacy, Regional Institute of Paramedical and Nursing Science, Aizawl, Mizoram, India
[5] Department of Biotechnology, Pachhunga University College, Aizawl, Mizoram, India

Corresponding authors
Esther Lalnunmawii,
essiezd.mzu@gmail.com
Zothanpuia, jpahnamte6@gmail.com

## ABSTRACT

Multidrug-resistant bacteria pose an alarming global health threat. The persistent rise in antibiotic-resistant infections continues to challenge healthcare systems worldwide. This study aims to evaluate the antimicrobial potential of 18 endophytic actinobacteria isolated from *Mirabilis jalapa* (L.) against clinically relevant multidrug-resistant pathogens. Among these, strain MIRK71 exhibited the strongest activity in secondary antimicrobial screening and was selected for further investigation. Molecular characterization using 16S rRNA gene sequencing identified MIRK71 as a *Streptomyces* species. Morphological analysis via scanning electron microscopy (SEM) revealed branched, filamentous colonies with microspore chains. Antimicrobial compounds were extracted from the culture filtrate grown in ISP1 broth using methanol under reduced pressure in a rotary evaporator. Gas chromatography-mass spectrometry (GC-MS) analysis detected 20 volatile compounds. Further profiling was conducted using high-performance thin-layer chromatography (HPTLC) with three optimized solvent systems: ethyl acetate:methanol:water (20:3:2) for glycosides (SS1), cyclohexane:ethyl acetate:formic acid (4:6:1) for phenols (SS2), and toluene:ethyl acetate:methanol:acetic acid (3:5:1:0.5) for anthracenes (SS3). Five peaks were recorded in SS1 and SS3, while seven peaks were observed in SS2 at 254 nm. Overall, this study contributes to the growing body of knowledge on endophytic actinobacteria and underscores their potential applications in antimicrobial therapy.

## INTRODUCTION

Endophytes are beneficial microorganisms that live within plants, contributing positively to their health without causing any diseases (*Wippel, 2023*). Few endophytes are known
to share compounds produced by the host plant (*Mishra, Passari & Singh, 2016*). This indicates that endophytes are present within plant tissues and subsist as reservoirs of bioactive metabolites (*Tsipinana et al., 2023*). Plants are widely used as traditional medicine in several parts of the world because they contain various potential substances for the treatment of chronic and infectious diseases. The use of medical plants as traditional medicines is particularly prominent in rural areas of many developing countries (*Aziz et al., 2018*). The endophytic bacteria benefit plants and their mutual association improves health or fitness of the plants with the help of various mechanisms like the induction of systematic resistance (*Liu et al., 2019*; *Purushitham et al., 2020*). The secondary metabolites produced by endophytes significantly contribute to the phytochemical compounds of medicinal plants (*Silva et al., 2020*). *Mirabillis jalapa* (Nyctaginaceae) is a well-known traditional medicinal plant and is used as folklore remedies in almost all parts of the world. The plant contains alkaloids, glycosides, saponins, flavonoids, tannins, lignin, phenols, and carbohydrates as its active constituents (*Saha, Deb & Deb, 2020*). In Mizoram, Northeast India, it is reported to treat certain ailments such as malaria, retained placenta, typhoid, anti-inflammatory, *etc.* (*Lalrinkima, 2013*; *Lalzarzovi & Lalramnghinglova, 2016*).

In addition, endophytes also secrete hybrid compounds from the host plant as a result of the symbiotic relationship with the host plant. Several bioactive compounds such as antibiotics are produced by endophytes, for instance, *Streptomyces sp.* NRRL 30562, an endophyte extracted from *Kennedia nigriscans*, has been noted for its production of actinomycin X2, a versatile polypeptide antibiotic that may be effective against human pathogenic bacteria (*Castillo et al., 2006*). Eventhough, antibiotics have saved millions of lives and significantly improved life expectancy over the last century, the emergence of multi-drug resistant (MDR) pathogens has threatened the clinical efficacy of many existing antibiotics (*Ahmed et al., 2024*). Overuse of antibiotics and over prescription has become the major factor for the emergence and dissemination of multi-drug-resistant strains of several groups of microorganisms (*Muteeb et al., 2023*). The widespread and, at times, unregulated use of antibiotics has resulted in the emergence of resistance to traditional antimicrobial agents among various bacterial pathogens, posing a significant public health challenge. For instance, 75% of a total *Staphylococcus aureus* isolated from clinical patients were MRSA (methicillin resistant *Staphylococcus aureus)* (*Gurung, Maharjan & Chhetri, 2020*). Researchers are increasingly turning their attention to herbal products in search of new leads to develop better drugs against MDR strains, as all this has resulted in severe consequences, including increased cost of medicines and patient mortality (*Gurung, Maharjan & Chhetri, 2020*). With the increase in the usage of antimicrobials, the complexities on which the bacterial pathogens exhibit the resistance mechanisms has increased. Scientists are struggling to control infections, but the development of new antimicrobial agents is not keeping pace with the rate of increasing resistance, as microorganisms evolve to possess better resistance mechanisms (*Krause, 1992*). Endophytes are the potential source to combat problem cause by Multi drug resistant organisms, among the endophytes, endophytic actinobacteria are a promising source for novel antimicrobial compounds. Actinomycetota are high guanine and cytosine ($\geq$55%) containing bacteria and it is one of the dominant phyla of the bacteria found on almost natural substrates

(*Hopwood, 2019*). These are important sources for novel antibiotics, hence having a high pharmacological and commercial interest in controlling infectious diseases (*Alam et al., 2022*). The majority of natural antibiotics that we know of are produced by actinobacteria. Actinobacteria notably produced at least two thirds of all known antibiotics consumed in the clinic today (*Van der Meij et al., 2017*; *Van Bergeijk et al., 2020*). Of the 20,000 biologically active compounds sourced from microorganisms (*Demain & Sanchez, 2009*), the majority are derived from actinobacteria accounting for 45% of all antibiotics currently in use (*Van der Meij et al., 2017*). It has been reported that 100,000 secondary metabolites are produced by *Streptomyces,* which is 70–80% of all natural bioactive products with pharmacological or agrochemical applications (*Abdel-Razek et al., 2020*). The present study aims to explore the traditional medicinal plants of Mizoram, Northeast India which falls under Indo-Burma biodiversity hotspots for the isolation and characterization of potential endophytic actinobacteria which can inhibit MDR human clinical pathogens.

## MATERIALS AND METHOD

**Collection and preparation of plant samples:** The whole plant (root, leaves and stem) of *Mirabilis jalapa* (L). was collected from Tanhril, Aizawl, Mizoram and was transferred to laboratory using sterile airtight plastic container. The whole plant was washed in running tap water for around 10 min and was dried under room temperature for overnight in the laboratory excluding direct sun light at room temperature and cut into small bits (1.0 × 0.5 cm).

**Isolation and morphological identification of endophytic actinobacteria:** The small cut bits were surfaced sterilised by immersion in 70% (v/v) ethanol for 60 s, followed by sodium hypochlorite solution (1% w/v, available chlorine) for 120 s and finally in 70% ethanol (v/v) for 30 s and then 4–5 bits were placed on ISP7 (Tyrosine agar) and SCA (Starch Casein Agar) media (*Kuster & Williams, 1964*). Nalidixic acid and cycloheximide (50 µg/ml) were used to supress the bacterial and fungal growth in the media (*Assad et al., 2021*). The plates were incubated at 28 ± 2 °C for 3–5 days. Individual colonies with characteristics of actinobacteria morphology were isolated and pure culture of the respective isolates were obtained by repeated streaking on ISP7 plates. The nature of the colony, the colour of aerial and substrate mycelium was studied (*Goodfellow & Haynes, 1984*). The spore chain morphologies of the isolates were studied using a scanning electron microscope (SEM). The mycelium structures were observed using a phase contrast microscope (Olympus), and the organisms were identified based on their shape and size, Gram reaction, and other pertinent morphological features according to Bergey's Manual of Determinative Bacteriology 9th edition (*Zothanpuia et al., 2018*).

### Primary screening

Actinobacteria isolates were tested for production of antimicrobial compounds against clinical pathogens (Table 1). The isolates were grown in tryptone yeast extract broth (ISP1) media in shaker incubator at 150 rpm for three days. The broth was centrifuged and the supernatant was collected which was further used for primary antimicrobial screening following agar well diffusion method (*Saadoun & Muhana, 2008*). Wells with

**Table 1   Test organisms (MDR pathogens) used for the antimicrobial screening.**

| S.No | Pathogens | Organisms | Resistance |
|---|---|---|---|
| 1. | 17 | *Escherichia coli* | Meropenem, Ciprofloxacin, Trimethoprim, Amoxicillin, Piperacillin, Cefuroxime, Ceftriaxone, Cefoperaxone, Cefepime, Ertapenem, Imipenem |
| 2. | 31 | *Escherichia coli* | Meropenem,Amikacin, Gentamicin, Ciprofloxacin,Trimethprim, Amoxicillin, Piperacillin, Cefuroxime, Ceftriaxone, Cefoperaxone, Cefepime, Ertapenem, Imipenem |
| 3. | 038 | *Staplylococcus aureus* | Vancomycin, Benzylpenicillin, Oxacillin, Ciprofloxacin, Erythromycin, Levofloxacin |
| 4. | 161 | *Staplylococcus aureus* | Vancomycin, Benzylpenicillin, Oxacillin, Ciprofloxacin, Erythromycin, Levofloxacin |
| 5. | 363 | *Enterococcus faecalis* | Daptomycin, Teicoplanin, Vancomycin, Tetracyclin, Nitrofurantoin, Benzylpenicillin, Gentamicin, Ciprofloxacin, Erythromycin, Levofloxacin |
| 6. | 365 | *Staplylococcus aureus* | Vancomycin, Benzylpenicillin, Oxacillin, Ciprofloxacin, Levofloxacin |
| 7. | 452 | *Staplylococcus aureus* | Vancomycin, Benzylpenicillin, Oxacillin, Ciprofloxacin, Erythromycin, Levofloxacin |
| 8. | 536 | *Enterococcus faecalis* | Vancomycin, Benzylpenicillin, Gentamicin, Daptomycin, Teicoplanin, Tetracycline, Ciprofloxacin, Erythromycin, Levofloxacin |
| 9. | 622 | *Staplylococcus aureus* | Vancomycin, Benzylpenicillin, Oxacillin, Ciprofloxacin, Erythromycin, Levofloxacin |
| 10. | 645 | *Staplylococcus aureus* | Rifampicin, Vancomycin, Benzylpenicillin, Oxacillin, Ciprofloxacin, Erythromycin, Levofloxacin |
| 11. | 721 | *Enterococcus faecalis* | Daptomycin, Teicoplanin, VaFfigncomycin, Tetracyclin, Nitrofurantoin, Benzylpenicillin, Erythromycin, |
| 12. | 790 | *Staplylococcus aureus* | Vancomycin, Benzylpenicillin, Oxacillin, Ciprofloxacin, Erythromycin, Levofloxacin |
| 13. | 809 | *Staplylococcus aureus* | Vancomycin, Benzylpenicillin, Oxacillin, Ciprofloxacin, Erythromycin, Levofloxacin |
| 14. | 880 | *Enterococcus faecalis* | Daptomycin, Teicoplanin, Vancomycin, Tetracyclin, Benzylpenicillin, Gentamicin, Ciprofloxacin, Erythromycin, Levofloxacin |
| 15. | 924 | *Staplylococcus aureus* | Rifampicin, Benzylpenicillin, Oxacillin, Ciprofloxacin, Erythromycin, Levofloxacin |
| 16. | 995 | *Klebsiella pneumoniae* | Meropenem, Gentamicin, Ciprofolxacin, Tigecycline, Trimethoprim, Amoxicillin, Piperacillin, Cefuroxime, Ceftriaxone, Cefoperaxone, Cefepime, Ertapenem, |
| 17. | 1097 | *Enterococcus faecalis* | Daptomycin, Teicoplanin, Vancomycin, Tetracyclin, Nitrofurantoin, Benzylpenicillin, Gentamicin, Ciprofloxacin, Erythromycin, Levofloxacin |
| 18. | 1098 | *Staplylococcus aureus* | Vancomycin, Benzylpenicillin, Oxacillin, Ciprofloxacin, Erythromycin, Levofloxacin |
| 19. | 1219 | *Enterococcus faecalis* | Vancomycin, Tetracycline, Erythromicin |
| 20. | 1246 | *Staphylococcus aureus* | Vancomycin, Benzylpenicillin, Oxacillin, Ciprofloxacin, Erythromycin, Levofloxacin |

**Table 1** (*continued*)

| S.No | Pathogens | Organisms | Resistance |
|------|-----------|-----------|------------|
| 21. | 1258 | *Enterococcus faecalis* | Daptomycin, Teicoplanin, Benzylpenicillin, Ciprofloxacin, Erythromycin, Levofloxacin Vancomycin, Tetracycline |
| 22. | 1266 | *Staplylococcus aureus* | Vancomycin, Benzylpenicillin, Oxacillin, Ciprofloxacin, Erythromycin, Levofloxacin |
| 23. | 1344 | *Staplylococcus aureus* | Vancomycin, Benzylpenicillin, Oxacillin, Ciprofloxacin, Erythromycin, Levofloxacin |
| 24. | 1379 | *Staplylococcus aureus* | Vancomycin, Benzylpenicillin, Oxacillin, Ciprofloxacin, Erythromycin, Levofloxacin |
| 25. | 1458 | *Staplylococcus aureus* | Vancomycin, Benzylpenicillin, Oxacillin, Ciprofloxacin, Erythromycin, Levofloxacin |
| 26. | 1468 | *Staplylococcus aureus* | Benzylpenicillin, Oxacillin, Ciprofloxacin, Levofloxacin |
| 27. | 1536 | *Staplylococcus aureus* | Vancomycin, Benzylpenicillin, Oxacillin, Ciprofloxacin, Erythromycin, Tetracycline, Levofloxacin |
| 28. | 2822 | *Staphylococcus aureus* | Teicoplanin, Vancomycin, Tetracycline, Trimethoprim, Benzylpenicillin, Oxacillin, Erythromycin |
| 29. | 2982 | *Staplylococcus aureus* | Vancomycin, Benzylpenicillin, Oxacillin, Ciprofloxacin, Levofloxacin |
| 30. | 2997 | *Escherichia coli* | Meropenem, Gentamicin, Ciprofolxacin, Trimethoprim, Amoxicillin, Piperacillin, Cefuroxime, Ceftriaxone, Cefoperaxone, Cefepime, Ertapenem, Imipenem, Amikacin |
| 31. | 3001 | *Staplylococcus aureus* | Vancomycin, Benzylpenicillin, Oxacillin, Ciprofloxacin, Erythromycin, Levofloxacin |
| 32. | 3054 | *Pseudomonas aeruginosa* | Gentamicin, Cyprofloxacin, Levofloxacin, Colistin, Cefeazidime, Aztreonam, Imipenem, Meropenem, Amikacin |
| 33. | 3069 | *Staplylococcus aureus* | Vancomycin, Benzylpenicillin, Oxacillin, Ciprofloxacin, Erythromycin |

**Notes.**

*Clinical pathogens were collected from Synod hospital, Durtlang, Aizawl, Mizoram.

6 mm diameter were made on the agar plate using sterile cork borer. The test organism was swabbed on the agar surface and 100 µl of the supernatant was added in the wells. The plates were allowed to stand for few minutes and incubated at 37 °C without inverting for 24 h. Clinical pathogens were collected from patients in Synod Hospital, Durtlang, Aizawl, Mizoram. Numbering system was used for marking the pathogens in order to make the patient anonymous. The clinical pathogen was isolated and identified using Vitek®MS (bioMérieux, Marcy l'Etoile, France) GP colorimetric identification card and the antibiotic susceptibility was determined using Vitek®MS AST (bioMérieux, Marcy l'Etoile, France).

**Extract preparation and antimicrobial screening:** The actinobacteria that was selected was inoculated for submerged fermentation. This process was carried out in 700 mL of ISP1 medium with a pH of 7.2, in a 500 mL conical flask under sterile conditions. The flasks were placed on a flask shaker at a speed of 120 rpm at room temperature for one week. After one week of fermentation, the medium was observed to become turbid.. The culture was harvested and centrifuged to remove cells and debris and the resultant broth was mixed with equal volume of solvents (methanol, ethyl acetate and dichloromethane) 1:1 ratio (v/v) to extract secondary metabolites (*Selvameenal, Radhakrishnan & Balagurunathan, 2009*).

After 24 h, the solvents were filtered out using Whatman filter paper. The excess solvent from the filtrated was evaporated under reduced pressure using a rotary evaporator to give crude extract. The extracts were subjected to antimicrobial screening. A concentration of 20 mg/ml of the crude extract was prepared for antimicrobial screening using agar well diffusion method (*Saadoun & Muhana, 2008*). A total of 4% methanol was used as a negative control. Small wells with a six mm diameter were created on an agar plate using sterile cork borers. The test organism was swabbed onto the agar surface, and 100 μL of a 20 mg/ml crude extract was added to the wells. The plates were then left to stand for a few minutes and were subsequently incubated at 37 °C without being inverted for 24 h.

**Identification of potential endophytic actinobacteria:** Genomic DNA was isolated and purified using a DNA extraction kit (Invitrogen) according to the manufacturer's protocol. Ribosomal RNA (16S rRNA) genes were amplified using universal bacterial primers 16SP1 and 16SP2 as forward and reverse primer (forward 16 s rRNA primer 5′-GTGCCAGCAGCCGCGG-3′ and reverse 16 s rRNA primer 5′-TACGGYTACCTTGTTACDACTT-3′). The reactions and conditions of the PCR were performed in a Veriti thermal cycler (Applied Biosystem, Singapore) in a total volume 25μl and the PCR conditions are as follows: initial denaturation step at 94 °C for 5 min, followed by 35 cycles of denaturation at 94 °C for 1 min, annealing at 62 °C for 1 min and extension at 72 °C for 1.2 min with a final extension step at 72 °C for 5 min and sequencing was done commercially at Eurofins Pvt. Ltd. Bangalore, India. Sequences were compared with the reference strains of actinobacteria from the NCBI genomic database using a BLASTn search tool to determine similarity percentages. The strains with the highest similarity percentages were obtained from the NCBI database.

## Phylogenetic analysis

The sequences of the 16S rRNA gene were compared with the NCBI database using BlastN and the most similar match sequence was selected. The sequences were aligned with pair wise alignment and multiple alignment using the program Clustal W packaged in the MEGA 11 software. From this data, a phylogenetic tree was constructed using the neighbor-joining tree method (*Keklik, 2023*). Bootstrap analysis was performed with MEGA 11 using Tamura-Nei (TN93).

**Analysis of the chemical components of the extract by Gas chromatography-mass spectrometry (GCMS):** Gas chromatography-mass spectrometry (GC-MS) was utilized to identify the volatile organic compounds (VOCs) in the methanolic extracts of *Streptomyces* sp.For GC–MS, the Clarus 690 Perkin Elmer GC, which was coupled with a mass detector Turbomass gold 5.1 spectrometer, and an Elite 1 (100% Dimethyl poly siloxane) capillary column measuring 123.5 MX 678 M. The instrument was initially set to a temperature of 40 °C and then ramped up to 150 °C at 10 °C/min and held for 5 min. After that the oven temperature was increased to 250 °C at a rate of 40 °C/min and maintained for eight minutes. The injection port temperature was set at 250 °C, while the helium flow rate was maintained at 1.5 mL/min. The ionization voltage used was 70 eV, and the samples were injected in a 10:1 split mode. The mass spectral scanning range was ser to 500–800 (m/z), and the ion source temperature and interface temperature were kept at 230 °C and 240 °C,

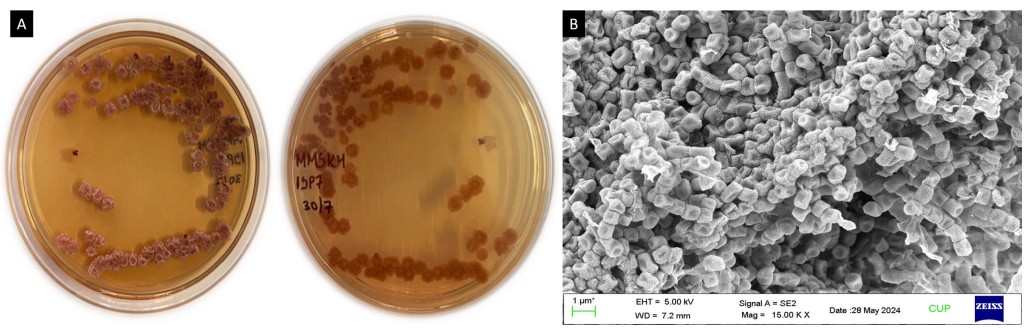

**Figure 1** (A) Morphological appearance of *Streptomyces* sp. strain MIRK71 and (B) SEM.
(A) *Streptomyces* sp. strain MIRK71 morpholpgy in ISP7 medium, (B) spore chain morphology of
*Streptomyces* sp. strain MIRK71.

respectively. The MS start time was three minutes and the end time was 31 min, with a solvent cut time of three minutes. The spectra of volatile compounds detected through GC-MS were compared and matched with NIST17 online library Ver.2.339

**Separation of compounds using high-performance thin-layer chromatography (HPTLC):** The extract was further analysed using CAMAG HPTLC instrument equipped with winCAT software. A stock solution of 50 mg/ml in distilled water was prepared. TLC plates with 0.2 mm precoated silica gel 60F$_{254}$ (Merck, Germany) of 2 × 10 cm was taken. The sample was spotted using Linomat 5 automated sample spotter (CAMEG) using 100 µl of syringe (Hamilton, Bonaduz, Switzerland). A phenolic, glycosides and mobile phase was used for profiling. Nitrogen gases were supplied on plates for simultaneous drying of bands. The extract loaded plates were kept in TLC twin trough developing chamber with the mobile phase Toluene: Ethyl acetate: Acetic Acid (2:4:0.1). The developed plate was dried and scanned using by TLC scanner with Wincats software. The plates were directly visualized after drying and finger print profile was documented under 254 nm and 366 nm in UV visible light. The peak display, peak densitogram and peak table was recorded

## RESULTS

### Isolation of endophytic actinobacteria

A total of 18 isolates (MIRK54, MIRK55, MIRK56…MIRK71) were obtained from *Mirabilis jalapa* Ten isolates were obtained from the roots, five isolates from the leaves and three isolates from the stem. ISP7 and SCA were used for the isolation process, 12 isolates were recovered from ISP7 media and six isolates were recovered from SCA. This clearly indicates that ISP7 is more suitable medium for the isolation of endophytic actinobacteria from *Mirabilis jalapa.* MIRK71 actinobacterial isolate has white colonies, aerial mass was white while the substrate mycelium was dark red and little pigments were produced on the media (Fig. 1A). A smooth spore surface with branched, filamentous and microspore chain colonies was identified by scanning electron microscope (Fig. 1B).

### Primary screening

The primary screening of antimicrobial was initially done for all the 18 isolates using supernatant of three days old broth culture grown in ISP1 media against 33 MDR clinical pathogens namely MRSA, VRE, *Klebsiella pneumoniae*, *Pseudomonas aeruginosa* and *Escherichia coli*. Out of the 18 isolates, MIRK55, MIRK68, MIRK69 and MIRK71 showed inhibition of pathogens growth to at least five or more of the test organisms with an inhibition zone ranging from 5–18 mm in diameter with the best inhibition shown by MIRK71.

### Extract preparation and secondary screening

Based on the primary screening, the best four isolates *i.e* MIRK55, MIRK68, MIRK69 and MIRK71 were subjected for extract preparation and secondary screening. Methanol, ethyl acetate and dichloromethane were used as a solvent for each isolate and a total of 12 extract were prepared and subjected to secondary screening using Agar well diffusion method. The extracts were tested only against 23 MDR clinical pathogens that showed positive results from the primary screening, including MRSA, VRE and *Pseudomonas aeruginosa*. Of the 12 extracts, methanolic crude extract of MIRK71 showed the best antimicrobial activity by inhibiting 18 pathogens with zone of inhibition ranging from 3–10 mm in diameter. Results of secondary antimicrobial screening suggest that the most commonly inhibited bacterial growth for methanolic extract of actinobacteria includes MRSA and VRE. Consequently, methanolic crude extract of MIRK71 was selected for further detailed characterization.

### Molecular identification

The best isolate MIRK71 was subjected to molecular identification using 16S rRNA gene sequencing, the isolate MIRK71 was identified as *Streptomyces* sp. that share 99.3% similarity with *Streptomyces* sp. NSC30 (MG833381). The sequence was deposited in NCBI GenBank database with an accession number OR896092.

### Phylogenetic analysis

16S rRNA gene sequences of MIRK71 and the most similar match sequences were used for the construction of phylogenetic tree. The phylogenetic tree was constructed using neighbor-joining method and the bootstrap analysis was performed using Tamura nei model as given in Fig. 2.

### Determination of volatile compounds using GC-MS

The volatile organic compounds (VOCs) was determined using Gas chromatography–mass spectroscopy (GC-MS). The methanolic crude extract of *Streptomyces* sp. MIRK71 revealed 20 VOCs within the retention time of 15–29 min (Table 2). The area percentage ranges from 10.111 (1H-indole-3-acetic acid, 5-chloro-2-methyl-1-(trimethylsilyl) silane) to 0.835% (Perylene). Of the 20 VOCs, most of the compounds were found to have antimicrobial activity.

### HPTLC profiling of the potential extract

Three optimized solvent system for mobile phase namely ethyl acetate: methanol: water (20:3:2), cyclohexane: ethyl acetate: formic acid (4:6:1) and toluene: ethyl acetate: methanol:

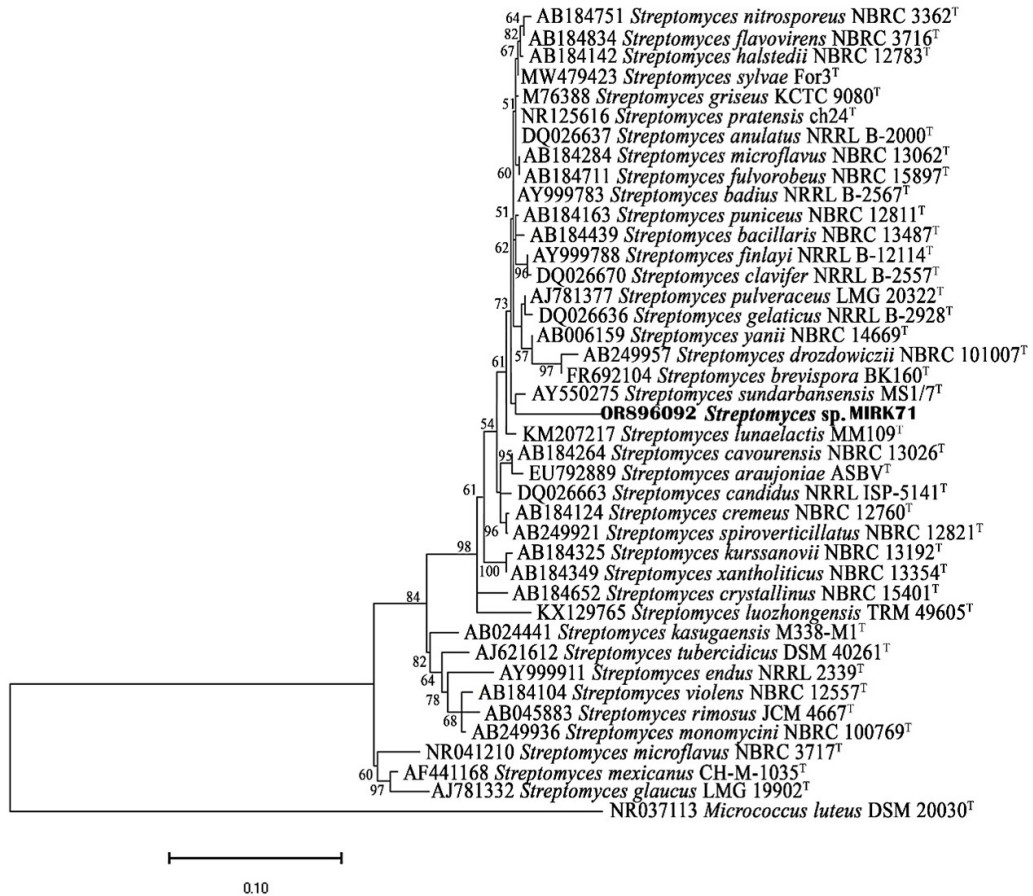

64 ┌ AB184751 *Streptomyces nitrosporeus* NBRC 3362ᵀ
82 │ AB184834 *Streptomyces flavovirens* NBRC 3716ᵀ
67 └ AB184142 *Streptomyces halstedii* NBRC 12783ᵀ
MW479423 *Streptomyces sylvae* For3ᵀ
5 │ M76388 *Streptomyces griseus* KCTC 9080ᵀ
NR125616 *Streptomyces pratensis* ch24ᵀ
DQ026637 *Streptomyces anulatus* NRRL B-2000ᵀ
AB184284 *Streptomyces microflavus* NBRC 13062ᵀ
60 │ AB184711 *Streptomyces fulvorobeus* NBRC 15897ᵀ
AY999783 *Streptomyces badius* NRRL B-2567ᵀ
51 ┌ AB184163 *Streptomyces puniceus* NBRC 12811ᵀ
62 ├ AB184439 *Streptomyces bacillaris* NBRC 13487ᵀ
│ AY999788 *Streptomyces finlayi* NRRL B-12114ᵀ
96 ┤ DQ026670 *Streptomyces clavifer* NRRL B-2557ᵀ
73 │ AJ781377 *Streptomyces pulveraceus* LMG 20322ᵀ
└ DQ026636 *Streptomyces gelaticus* NRRL B-2928ᵀ
AB006159 *Streptomyces yanii* NBRC 14669ᵀ
61 │57 ┌ AB249957 *Streptomyces drozdowiczii* NBRC 101007ᵀ
97 └ FR692104 *Streptomyces brevispora* BK160ᵀ
AY550275 *Streptomyces sundarbansensis* MS1/7ᵀ
**OR896092  Streptomyces sp. MIRK71**
54 ├ KM207217 *Streptomyces lunaelactis* MM109ᵀ
95 │ AB184264 *Streptomyces cavourensis* NBRC 13026ᵀ
└ EU792889 *Streptomyces araujoniae* ASBVᵀ
61 ┌ DQ026663 *Streptomyces candidus* NRRL ISP-5141ᵀ
│ AB184124 *Streptomyces cremeus* NBRC 12760ᵀ
96 ┤ AB249921 *Streptomyces spiroverticillatus* NBRC 12821ᵀ
98 │ ┌ AB184325 *Streptomyces kurssanovii* NBRC 13192ᵀ
100 ┤ AB184349 *Streptomyces xantholiticus* NBRC 13354ᵀ
├ AB184652 *Streptomyces crystallinus* NBRC 15401ᵀ
└ KX129765 *Streptomyces luozhongensis* TRM 49605ᵀ
84 │ AB024441 *Streptomyces kasugaensis* M338-M1ᵀ
82 ├ AJ621612 *Streptomyces tubercidicus* DSM 40261ᵀ
64 ├ AY999911 *Streptomyces endus* NRRL 2339ᵀ
78 ├ AB184104 *Streptomyces violens* NBRC 12557ᵀ
68 ├ AB045883 *Streptomyces rimosus* JCM 4667ᵀ
└ AB249936 *Streptomyces monomycini* NBRC 100769ᵀ
NR041210 *Streptomyces microflavus* NBRC 3717ᵀ
60 ┌ AF441168 *Streptomyces mexicanus* CH-M-1035ᵀ
97 └ AJ781332 *Streptomyces glaucus* LMG 19902ᵀ
NR037113 *Micrococcus luteus* DSM 20030ᵀ

0.10

**Figure 2  Neighbor joining phylogenetic tree using Tamura nei model based on 16SrRNA gene sequences.**

acetic acid (3:5:1:0.5) for glycosides (SS1), phenols (SS2) and anthracene (SS3) respectively were used at a volume of eight μl of methanolic extract performed at UV 254 nm and 366 nm in visualizer and visible light and single brown spots were detected (Fig. 3). The images were captured and scanned in CAMAG TLC Scanner using VisionCATS 3.2 SP2 software.

The peak display, peak densitogram, and peak table were recorded and analysed and Rf values were measured then the presence of positive result was recorded on 254 nm and 366 nm. five peaks were recorded in SS1 (Fig. 4) whereas, seven peaks were recorded in SS2 (Fig. 5) and five peaks were recorded in SS3 (Fig. 6) at 254 nm. The peaks with Rf value and area (%) of SS1, SS2 and SS3 are shown in Tables 3, 4 and 5 respectively.

## DISCUSSION

Microorganisms can develop multi-drug resistant when they acquire several resistance mechanisms against several drugs. Since MDR microbes are increasing, the development of new antimicrobial agents that are effective against resistant microbes is on high demand

**Table 2  Volatile organic compounds detected from the methanolic crude extract of MIRK71.**

| S.no | RT | Scan | Height | Area% | Name | Mol.Wt | Formula |
|---|---|---|---|---|---|---|---|
| 1 | 15.88 | 2,086 | 1247,748 | 0.835 | Perylene | 525 | C20H12 |
| 2 | 23.052 | 2,109 | 15,679,657 | 7.478 | Spiro (2,4) Hept-S-ene, S-trimethyl-1-trimethysilyl- | 252 | C14H28Si2 |
| 3 | 23.592 | 2,195 | 12,180,329 | 10.111 | 1H-indole-3-acetic acid, 5-chloro-2-methyl-1-(trimethylsilyl) silane | 324 | C20H28Si2 |
| 4 | 23.78 | 2,204 | 3,311,306 | 2.640 | N- decanoic acid | 172 | C10H20O2 |
| 5 | 24.09 | 2,211 | 2,038,519 | 0.845 | Anthracene, 9,10, dihydro-9,10-Bis (trimethylsilyl) | 324 | C20H28Si2 |
| 6 | 24.412 | 2,215 | 2,438,512 | 1.636 | Acethlydrazine, N2-{-4-(Thitan-3-yloxy) Benzylideno} | 250 | C12H14O2N2S |
| 7 | 24.792 | 2,281 | 1,541,942 | 1.457 | 1-Nonylcycloheptane | 224 | C16H32 |
| 8 | 24.953 | 2,319 | 2,176,142 | 2.303 | 1,3-Dioxolane, 2-pentadecyl- | 284 | C18H36O2 |
| 9 | 25.053 | 2,369 | 1,440,245 | 0.989 | 1,4-Naphthoquinone, 6-ethyl-2,3,5,7-tetrahydroxy- | 250 | C12H10O6 |
| 10 | 25.293 | 2,380 | 1,576,417 | 1.138 | Octanal, 7-methoxy-3-7-dimethyl- | 186 | C11H22O2 |
| 11 | 25.413 | 2,390 | 1,512,041 | 1.612 | Dinitrophenyl-L- isoleucylglycine | 368 | C15H20O7N4 |
| 12 | 25.813 | 2,414 | 1,038,761 | 0.898 | 3-pethanol-3, methyl | 284 | C18H36O2 |
| 13 | 25.913 | 2,424 | 1,354,133 | 0.882 | Anthracene, 9,10, dihydro-9,10-Bis (trimethylsilyl) | 324 | C20H28Si2 |
| 14 | 26.193 | 2,441 | 1,387,282 | 0.925 | Tetrasiloxane, decamethyl | 310 | C10H30O3Si4 |
| 15 | 26.403 | 2,456 | 1,402,452 | 1.525 | 2-phenyl-1-Benzimidazolyl) acetic acid | 252 | C15H12O2N2 |
| 16 | 26.50 | 2,474 | 1,281,008 | 1.684 | Anthracene, 9,10, dihydro-9,10-Bis (trimethylsilyl) | 324 | C20H28Si2 |
| 17 | 26.693 | 2,478 | 1,702,987 | 1.112 | Tetrasiloxane, decamethyl | 310 | C10H30O3Si4 |
| 18 | 26.803 | 2,591 | 1,477,066 | 0.921 | (2-Phenyl-1-Benzeimidazolyl) acetic acid | 252 | C15H12O2N2 |
| 19 | 27.143 | 2,648 | 1,135,466 | 1.107 | 1,3-Dioxolane, 2-pentadecyl- | 284 | C18H36O2 |
| 20 | 29.814 | 2,681 | 1,390,848 | 1.393 | Anthracene, 9,10, dihydro-9,10-Bis (trimethylsilyl) | 324 | C20H28Si2 |

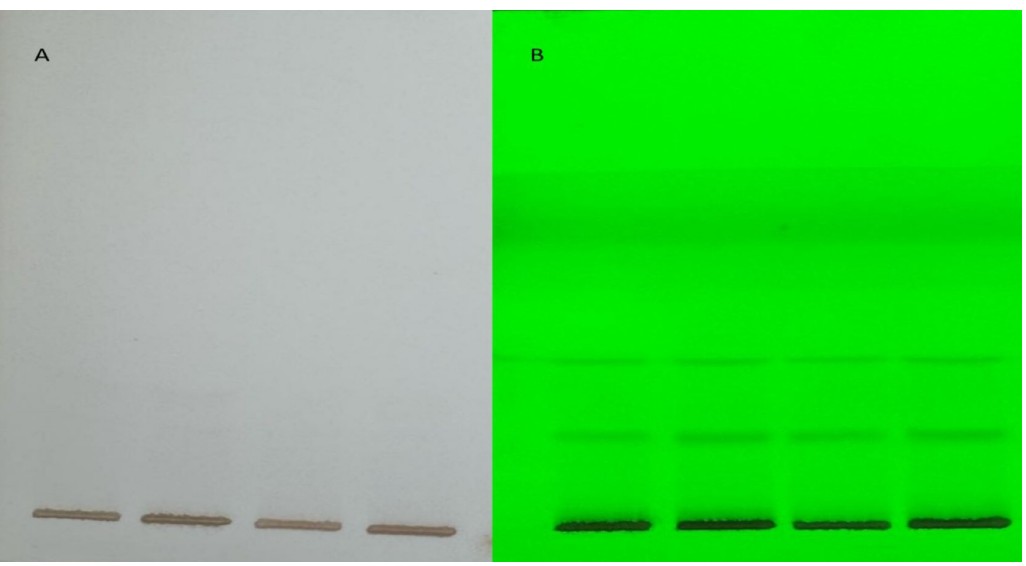

**Figure 3  Visible light and single brown spots detected, captured and scanned in CAMAG TLC Scanner using VisionCATS 3.2 SP2 software.**

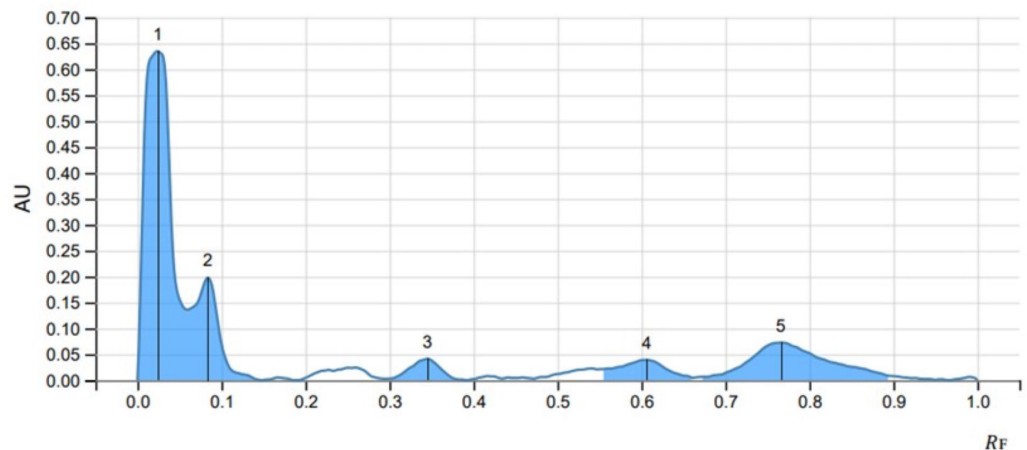

**Figure 4** HPTLC chromatogram of 8 μl methanolic extract in SS1 performed at UV 254.

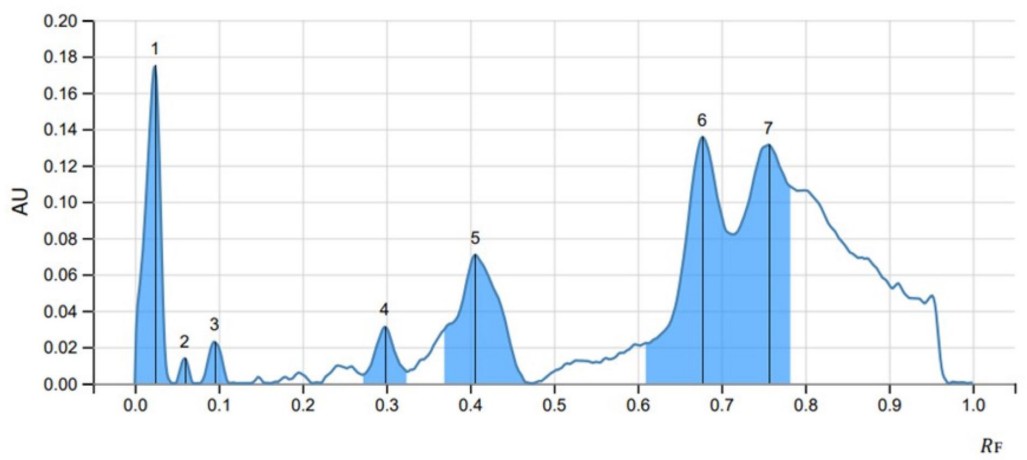

**Figure 5** HPTLC chromatogram of SS2 at 254 nm.

because this is the most effective way to combat resistant organisms (*Kenneth, 2009*). Natural products such as from traditional medicinal plants are in great demand since it has abundant biological properties and it could be a source for discovering novel bioactive compounds. Due to this reason the study of medicinal plants has an increasing interest, many research are based on medicinal plants as a potential source for the discovery of bioactive compounds such as anti-microbial compounds for the development of novel classes of antibiotics (*Schultes, 1960*).

*Mirabilis jalapa* has been recognized for their many pharmacological activities such as its anti-inflammatory (*Singh et al., 2010*) antioxidant (*Zachariah et al., 2011*; *Hajji et al., 2010*; *Oladunmoye, 2012*) and antibacterial (*Hajji et al., 2010*; *Cammue et al., 1992*) properties. It is a well-known traditional medicinal plant of Mizoram (*Lalzarzovi & Lalramnghinglova,*

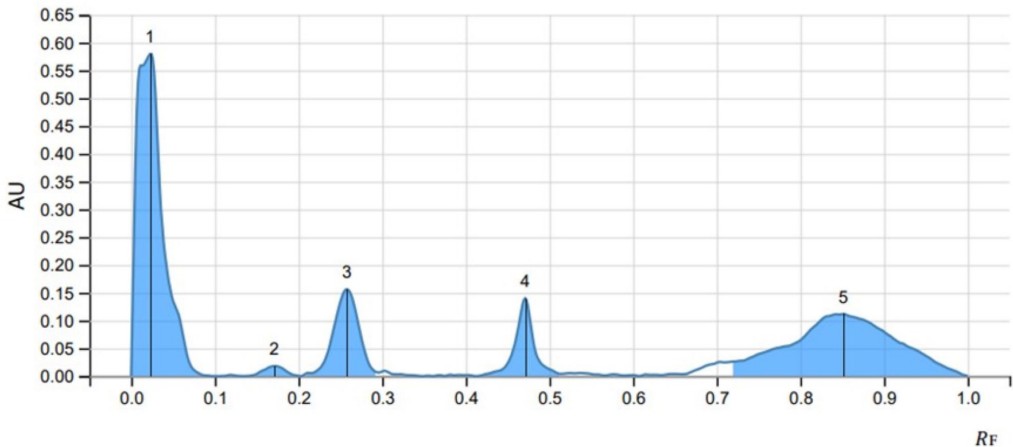

**Figure 6  HPTLC chromatogram of SS3 at 254 nm.**

**Table 3  HPTLC finger print profile for methanol extract of *Streptomyces* sp. MIRK71 in SS1 at 254 nm.**

| Peak | Start | | Max | | | End | | Area | |
|---|---|---|---|---|---|---|---|---|---|
| | $R_F$ | H | $R_F$ | H | % | $R_F$ | H | A | % |
| 1 | 0.000 | 0.0000 | 0.024 | 0.6345 | 64.35 | 0.060 | 0.1363 | 0.02380 | 55.03 |
| 2 | 0.060 | 0.1363 | 0.084 | 0.1979 | 20.08 | 0.147 | 0.0000 | 0.00700 | 16.18 |
| 3 | 0.295 | 0.0028 | 0.345 | 0.0415 | 4.21 | 0.384 | 0.0013 | 0.00173 | 4.00 |
| 4 | 0.555 | 0.0215 | 0.606 | 0.0394 | 3.99 | 0.661 | 0.0046 | 0.00265 | 6.14 |
| 5 | 0.669 | 0.0062 | 0.766 | 0.0727 | 7.37 | 0.895 | 0.0084 | 0.0084 | 18.65 |

*2016*) and due to this reason, it was selected for the present investigation. In the present study, 18 actinobacterial strains were isolated, using surface sterilization method using 70% ethanol and 1% sodium hypochloride as it can effectively reduce microbial contamination on surfaces. The ethanol step can reduce microbial contamination by 3–5 log10 CFU, while the sodium hypochlorite solution can reduce it by 4–6 log10 CFU. The combined efficacy of this protocol can lead to a reduction in microbial contamination by 6–8 log10 CFU or more, covering a broader spectrum of microorganisms and providing a longer-lasting effect (*Rutala & Weber, 2021*). Ten isolates were isolated from the roots, five from the leaves and three from the stem. Two different nutritional media (ISP7 and SCA) were used for the isolation since nutrient uptake could differ from one organism to another. The isolation results showed that ISP7 media yield better results than SCA which was in accordance with earlier studies (*Khirennas et al., 2023*).

Actinobacteria are well known for producing different types of bioactive compounds including antibiotics, antitumor and immunosuppressive agents, plant growth hormones, and biological substances like enzymes, alkaloids, and vitamins, which can play a vital role in the pharmaceutical industries (*Uzma et al., 2018*; *Wu et al., 2010*). Scanning electron microscopy showed the structure of aerial mycelia and spores. A similar report of branched,

**Table 4  HPTLC finger print profile for methanol extract of *Streptomyces* sp. MIRK71 in SS2 at 254 nm.**

| Peak | Start | | Max | | | End | | Area | |
|---|---|---|---|---|---|---|---|---|---|
| | $R_F$ | H | $R_F$ | H | % | $R_F$ | H | A | % |
| 1 | 0.000 | 0.0000 | 0.024 | 0.1748 | 30.14 | 0.045 | 0.0000 | 0.00361 | 14.21 |
| 2 | 0.048 | 0.0000 | 0.060 | 0.0137 | 2.36 | 0.069 | 0.0000 | 0.00015 | 0.60 |
| 3 | 0.077 | 0.0000 | 0.095 | 0.0228 | 3.93 | 0.113 | 0.0000 | 0.00041 | 1.61 |
| 4 | 0.273 | 0.0046 | 0.298 | 0.0310 | 5.35 | 0.324 | 0.0065 | 0.00087 | 3.44 |
| 5 | 0.369 | 0.0295 | 0.406 | 0.0710 | 12.19 | 0.469 | 0.0003 | 0.00400 | 15.73 |
| 6 | 0.605 | 0.0210 | 0.677 | 0.1356 | 23.38 | 0.713 | 0.0819 | 0.00782 | 30.73 |
| 7 | 0.713 | 0.0819 | 0.756 | 0.1313 | 22.65 | 0.790 | 0.1060 | 0.00857 | 33.69 |

**Table 5  HPTLC finger print profile for methanol extract of *Streptomyces* sp. MIRK71 in SS3 at 254 nm.**

| Peak | Start | | Max | | | End | | Area | |
|---|---|---|---|---|---|---|---|---|---|
| | $R_F$ | H | $R_F$ | H | % | $R_F$ | H | A | % |
| 1 | 0.000 | 0.0000 | 0.023 | 0.5796 | 57.65 | 0.097 | 0.0000 | 0.02143 | 44.65 |
| 2 | 0.134 | 0.0000 | 0.171 | 0.0180 | 1.79 | 0.203 | 0.0000 | 0.00054 | 1.12 |
| 3 | 0.206 | 0.0022 | 0.258 | 0.1563 | 15.54 | 0.295 | 0.0063 | 0.00548 | 11.42 |
| 4 | 0.413 | 0.0000 | 0.471 | 0.1397 | 13.90 | 0.513 | 0.0040 | 0.00353 | 7.35 |
| 5 | 0.715 | 0.0252 | 0.852 | 0.1118 | 11.12 | 1.000 | 0.0002 | 0.01701 | 35.45 |

filamentous and microspore chain colonies with smooth spore surface morphology has been reported by *Ullah et al. (2012)*.

From all the 18 isolates subjected to primary antimicrobial screening, four isolates with the best antimicrobial activity were subjected to extract preparation and secondary antimicrobial screening. The methanolic extract of isolate MIRK71 showed the best antimicrobial activity based on the secondary screening against clinical MDR pathogens collected from local hospital (Synod hospital, Durtlang, Aizawl, India) of Mizoram. Thirty-three MDR pathogens were collected and primary screening was done using 18 actinobacterial isolates against all the 33 MDR clinical pathogens (*Aravamuthan et al., 2010*) isolated actinobacteria from the soil and tested against clinical MDR pathogens and found antibacterial activity which supports the present investigation. The same test was done by *Singh et al. (2012)* from actinobacteria isolated from the soil and found antimicrobial activity against three human pathogenic bacteria *i.e.,* vancomycin-resistant *Enterococci* (VRE), methicillin-resistant *Staphylococcus aureus* (*S. aureus*) and *Escherichia coli* (*E. coli*). The identification of MIRK71 as *Streptomyces* sp. could explain the antimicrobial activity because the genus streptomyces is a well-known and one of the largest antibiotic-producing genera among the microbes which is evident in the previous studies (*Watve et al., 2001*; *Atta, 2015*).

The methanolic crude extract composition of *Streptomyces sp.* MIRK71 was determined by gas chromatography. This analysis was done mainly to check the volatile components of the methanolic crude extract of MIRK71. The compounds in the crude extract

were identified by comparing their retention times and relative retention factors with authentic standards. The percentage composition was evaluated based on peak areas using area normalization. The detected components such as Anthracene, decanoic acid (*Kitahara et al., 2004*; *Shen et al., 2021*), octanal (*Liu et al., 2012*), hydrazine (*Popiołek, 2021*), dioxolanes (*Küçük et al., 2011*; *Ovsyannikova et al., 2013*) and perylene were earlier known to have antimicrobial activity (*Debbab et al., 2012*; *Dardeer, Taha & Elamary, 2020*) studied anthracene as an antimicrobial agent. A study on perylene as an antimicrobial was done against *Mycobacterium tuberculosis* and *Staphylococcus aureus* (MRSA) by *Yılmaz et al. (2023)*. Other studies on perylene as an antimicrobial agent was also done by *Rostaminejad et al. (2024)* which supports our present investigation. HPTLC analysis is a conventional technique used widely for chemical profiling, identification, isolation and quantification of bioactive compounds (*Jones et al., 2007*). In the present study, HPTLC is used for the analysis of the methanolic crude extract of *Streptomyces sp.* three solvent system were used and a separation of different compounds which was clearly evident by producing a single brown spot in mobile phase was observed. Glycosides, phenols and anthracene solvent system were used and five peaks were observed in glycosides, seven peaks in Phenols and five peaks in anthracene was observed. This indicate that, in the methanolic crude extract of *Streptomyces sp.* glycosides, phenols and anthracene are present and many of the antimicrobial agent belongs to glycosides, phenols or anthracene (*El Malah et al., 2020*; *Takó et al., 2020*; *Dardeer, Taha & Elamary, 2020*). In the present investigation, we have employed crude extracts for our analysis because these extracts encompass a diverse range of bioactive compounds, reflecting the complexity of natural substances rather than isolating the effects of a single compound. However, utilizing crude extracts does come with its own set of challenges, particularly the inability to identify which specific compound is responsible for the exact biological activities. As a result, this study aims to lay the groundwork for future research focused on the isolation and purification of individual compounds, employing HPTLC as a potential methodology to achieve this goal.

## CONCLUSION

The present investigation has shown that actinobacteria with antimicrobial potential against MDR clinical pathogens was successfully isolated from *Mirabilis jalapa* (L). which is a well-known traditional medicinal plant of the Mizo. The methanolic crude extract of isolate MIRK71 was proved to have the best antimicrobial potential against clinical MDR pathogens. However further investigation is needed for isolation and quantification of antimicrobial compound present in the methanolic crude extract of *Strepto myces* sp. (MIRK71). All the above investigation indicated that the *Streptomyces* sp. MIRK71 could be a valuable source for isolating bioactive compounds that can be used as antibiotics in the pharmaceutical industry.

### Funding

This work was supported by the Department of Science and Technology, Science and Engineering Research Board (DST-SERB), Government of India, vide project sanction no.: EEQ/2022/000878; Indian Council of Medical Research (ICMR) under sanction no.: ECD/NER/5/2022-23; UGC SRG F.30-555/202t(BSR) and RPG (11/1-349/2022/FIN-B/). The funders had no role in study design, data collection and analysis, decision to publish, or preparation of the manuscript.

### Grant Disclosures

The following grant information was disclosed by the authors:
Department of Science and Technology, Science and Engineering Research Board (DST-SERB).
Government of India: EEQ/2022/000878.
Indian Council of Medical Research (ICMR): ECD/NER/5/2022-23.
UGC SRG: F.30-555/202t(BSR).
RPG: 11/1-349/2022/FIN-B/.

### Competing Interests

Zothanpuia is an Academic Editor for PeerJ.

### Author Contributions

- Lalrokimi conceived and designed the experiments, performed the experiments, analyzed the data, prepared figures and/or tables, authored or reviewed drafts of the article, and approved the final draft.
- Purbajyoti Deka conceived and designed the experiments, authored or reviewed drafts of the article, and approved the final draft.
- William Carrie conceived and designed the experiments, authored or reviewed drafts of the article, and approved the final draft.
- Lallawmsangi conceived and designed the experiments, authored or reviewed drafts of the article, and approved the final draft.
- Christine Vanlalbiakdiki Sailo performed the experiments, authored or reviewed drafts of the article, and approved the final draft.
- Lalrosangpuii performed the experiments, authored or reviewed drafts of the article, and approved the final draft.
- Felicia Lalremruati performed the experiments, authored or reviewed drafts of the article, and approved the final draft.
- Awmpuizeli Fanai performed the experiments, authored or reviewed drafts of the article, and approved the final draft.
- Yasangam Umbon performed the experiments, analyzed the data, authored or reviewed drafts of the article, and approved the final draft.

- Esther Lalnunmawii conceived and designed the experiments, performed the experiments, analyzed the data, authored or reviewed drafts of the article, and approved the final draft.
- Zothanpuia conceived and designed the experiments, performed the experiments, analyzed the data, prepared figures and/or tables, authored or reviewed drafts of the article, and approved the final draft.

## DNA Deposition

The following information was supplied regarding the deposition of DNA sequences:

The sequence is available at NCBI: OR896092.

## Data Availability

The data is available in the Supplementary File.

## Supplemental Information

Supplemental information for this article can be found online at http://dx.doi.org/10.7717/peerj.19683#supplemental-information.

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
