# Peer review of "Antibacterial potential and chromatographic profiling of bioactive compounds from endophytic Streptomyces sp. strain MIRK71 isolated from Mirabilis jalapa (L.)"

_PeerJ, doi:10.7717/peerj.19683_

## Round 0.1 · original submission · Minor Revisions

**Language Note:** PeerJ staff have identified that the English language needs to be improved. When you prepare your next revision, please either (i) have a colleague who is proficient in English and familiar with the subject matter review your manuscript, or (ii) contact a professional editing service to review your manuscript. PeerJ can provide language editing services - you can contact us at [email protected] for pricing (be sure to provide your manuscript number and title). – PeerJ Staff

·

Basic reporting

Nowadays, the search for bacteria that produce compounds with antimicrobial potential against multidrug-resistant bacteria is an important issue worldwide. The aim of the study is well-focused. The article is well written, although I must mention some minor typos and inaccuracies:
- The supplementary tables must be referred to in the manuscript. Figure 3 is not cited in the text and does not provide relevant information.
- In my opinion, Figures 1, 2, 3, 4, 5, and 6 should be included between the paragraphs of the text to improve the flow and comprehension of the study.
- The information on the phylum Actinobacteria mentioned in the introduction is a bit outdated. Since 2021, the proper name of the taxon is Actinomycetota (Oren and Garrity, 2021). For an extensive review of its physiology and natural compounds, see Barka et al., 2016.
- Line 2: Instead of Jalapa, better jalapa. The first letter of the specific epithet in a Latin name is not capitalized. This word should be corrected in the title.

Experimental design

- The heading of the supplementary tables is incomplete and does not describe what these data relate to. It could be modified as follows: “Primary antimicrobial screening of isolates against 33 clinical MDR pathogens. Diameter of the inhibition zone (in mm)”

- Tests for the production of antimicrobial compounds against clinical pathogens could be carried out in triplicate to obtain deviation standards. I understand that they were only done as a screening, but keep this in mind for future occasions.

Validity of the findings

-

Additional comments

The results are promising, and it would be interesting to continue the research to try to identify the compounds detected in the HPTLC chromatograms.

Reviewer 2 ·

Basic reporting

Clear and unambiguous, professional English used throughout - Yes
Literature references, sufficient field background/context provided.- yes
Professional article structure, figures, and tables. Raw data shared.-yes
Self-contained with relevant results to hypotheses.-yes

Experimental design

-

Validity of the findings

-

Additional comments

This research article discusses the antimicrobial effect of bioactive compounds of an endophytic Streptomyces sp. strain MIRK71, which is isolated from Mirabilis Jalapa (L).

Comments
Title: The title can be revised for better clarity, like
“Antibacterial potential and chromatographic profiling of bioactive compounds obtained from endophytic Streptomyces sp. Strain MIRK71 Isolated from Mirabilis jalapa (L.)”
Abstract: The abstract does not reveal the study. So, I think it is better to restructure the abstract, incorporating aims and objectives, materials and methods, results, and conclusion.

Introduction
The introduction is very vast and does not mention the aims and objectives of the study. so
Please structure the introduction, focusing on the aims and objectives of the study.

Materials and methods
In line 105, Isolation and morphological identification of endophytic actinobacteria:
What is the efficacy of this method for isolating bacteria, considering that you are using 70% (v/v) ethanol for 60 seconds, followed by a 1% (w/v) sodium hypochlorite solution (available chlorine) for 120 seconds, and finally treating with 70% (v/v) ethanol for 30 seconds?. Please incorporate the details in the discussion section.
In line 114, “The spore chain morphologies of the isolates were studied using a scanning electron microscope (SEM)O” Why did the authors use SEM to study spore arrangements? Is it possible to study the arrangement of spores using a 40X or 100X light microscope with 1000x magnification? Similarly, the mycelium was studied using a phase-contrast microscope. Can the authors specify the magnification of the microscope they used?
In line 117, the bacterial identification, the authors should be more specific about how they used "Bergey’s Manual of Determinative Bacteriology, 9th edition" to identify the bacteria. Providing more detail on this process would help readers better understand the identification procedure. Please do include the reference number for this.

Discussion
Please incorporate the strengths and weaknesses of the current study, since the authors have used crude extracts.

Please also add a brief description of the future directions of the study.

Reviewer 3 ·

Basic reporting

Overall, this article meets the standard basic reporting of the findings, however, there are a few recommendations that need to be highlighted:
Improvements in explaining the connection between one statement to another statement. For example, in the introduction section:
Line 91: Recommendation to link paragraph 1 to paragraph 3: such as, In addition, endophytes also secrete hybrid compounds from the host plant, and endophytes as a result of the symbiotic relationship. Give examples of compounds such as antibiotics, anti-inflammatories, and anti-biofilm.

Experimental design

All procedures were handled according to standard procedures.

Validity of the findings

-

---

## Round 0.2 · accepted · Accept

All three reviewers now appeared to be satisfied that the authors have addressed all previous concerns.

·

Basic reporting

No comments

Experimental design

No comments

Validity of the findings

No comments

Reviewer 2 ·

Basic reporting

Clear and unambiguous, professional English used throughout.

Experimental design

Original primary research within Aims and Scope of the journal.

Validity of the findings

Impact and novelty not assessed. Meaningful replication encouraged where rationale & benefit to literature is clearly stated

Additional comments

No comments

Reviewer 3 ·

Basic reporting

No comments.

Experimental design

No comments

Validity of the findings

The findings are shown as an original article and are valid for publication.